# Intradural Pediatric Spinal Tumors: An Overview from Imaging to Novel Molecular Findings

**DOI:** 10.3390/diagnostics11091710

**Published:** 2021-09-18

**Authors:** Antonio Marrazzo, Antonella Cacchione, Sabrina Rossi, Alessia Carboni, Carlo Gandolfo, Andrea Carai, Angela Mastronuzzi, Giovanna Stefania Colafati

**Affiliations:** 1Neuroradiology Unit, Imaging Department, Bambino Gesù Children’s Hospital, IRCCS, 00165 Rome, Italy; antonio.marrazzo@opbg.net (A.M.); alessia.carboni@opbg.net (A.C.); carlo.gandolfo@opbg.net (C.G.); 2Department of Paediatric Haematology/Oncology, Cell and Gene Therapy, Bambino Gesù Children’s Hospital, IRCCS, 00165 Rome, Italy; antonella.cacchione@opbg.net (A.C.); angela.mastronuzzi@opbg.net (A.M.); 3Pathology Unit, Department of Laboratories, Bambino Gesù Children’s Hospital, IRCCS, 00165 Rome, Italy; sabrina2.rossi@opbg.net; 4Neurosurgery Unit, Department of Neuroscience and Neurorehabilitation, Bambino Gesù Children’s Hospital, IRCCS, 00165 Rome, Italy; andrea.carai@opbg.net

**Keywords:** spine tumor, children, cancer predisposition syndromes

## Abstract

Pediatric spinal tumors are rare and account for 10% of all central nervous system tumors in children. Onset usually occurs with chronic nonspecific symptoms and may depend on the intra- or extradural neoplastic location. Meningiomas, schwannomas, and neurofibromas are the most common intradural-extramedullary lesions, while astrocytomas and ependymomas represent the majority of intramedullary tumors. The new molecular discoveries regarding pediatric spinal cancer currently contribute to the diagnostic and therapeutic processes. Moreover, some familial genetic syndromes can be associated with the development of spinal tumors. Currently, magnetic resonance imaging (MRI) is the standard reference for the evaluation of pediatric spinal tumors. Our aim in this review was to describe the imaging of the most frequent intradural intra/extramedullary pediatric spinal tumors and to investigate the latest molecular findings and genetic syndromes.

## 1. Introduction

Pediatric spinal tumors are relatively rare and account for 10% of all central nervous system tumors in children [1,2,3]. Related to location, spinal neoplasms are divided into extradural (two-thirds of cases), intradural-extramedullary, and intramedullary (one-third of cases) lesions [3]. Extradural neoplasms have a variable origin and arise from bone, soft tissues, and meningeal sheets; intradural-extramedullary neoplasms are usually meningiomas, schwannomas, and neurofibromas. Intramedullary cancers mainly include glial tumors such as astrocytomas and ependymomas, which represent 90% of all intramedullary neoplasms [4], but can also include gangliogliomas and hemangioblastomas.

Onset symptoms are often non-specific and include variable pain, loss of balance, torticollis, progressive scoliosis, motor regression, and hydrocephalus [5]. Typical sensory symptoms are linked to the involvement of the sensory fibers and sometimes represent the onset of ependymomas due to their central location and their proximity to the spinothalamic tract. On the other hand, deficit sphincter symptoms are related to a low spinal medullary localization and can be caused by myxopapillary ependymoma due to its typical lumbosacral location [1].

Magnetic resonance imaging (MRI) is the standard reference for the evaluation of pediatric spinal tumors because it allows (i) neuraxis panoramic evaluation, (ii) detailed anatomical study of the region of interest, (iii) radiological characterization of the tumor and diagnostic hypothesis. An MRI evaluation includes pre- and post-contrast T1-weighted sequences, T2-weighted sequences, fat suppression sequences, “detailed study” with heavily T2-weighted MR “cisternography” sequences, diffusion-weighted imaging (DWI), and susceptibility-weighted imaging (SWI). Advanced neuroimaging methods, such as diffusion tensor imaging (DTI), perfusion technique, spectroscopy, and functional imaging, are not yet included in a “standard” spinal cord evaluation.

Computed tomography (CT) and radiographic imaging can be useful to evaluate bone involvement and spinal stability in selected cases.

The advancement of molecular findings in pediatric central nervous system tumors (CNS) provides additional information regarding the tumor subtypes and their biological behaviors and patient outcomes. These novel findings help to reach a detailed diagnosis, obtaining new therapeutic options by targeted drugs. These molecular advances have also been described for spinal neoplasms.

Our aim in this review was to describe the imaging findings of the most frequent intradural intra/extramedullary pediatric spinal tumors according to the World Health Organization (WHO) 2016 revised classification [6] and to investigate the connection with the latest molecular findings and genetic syndromes. It is also interesting to mention some very rare spinal tumors that can be exceptionally diagnosed in children, such as the diffuse leptomeningeal glioneuronal tumor (DL-GNT), due to its typical radiological aspect.

## 2. Gliomas and Mixed Neuronal–Glial Tumors

### 2.1. Gliomas

Astrocytomas are the most common pediatric intramedullary spinal tumors (approximately 60%) [1,2,7] and are frequently found in the first decade of life [3]. Among pediatric low-grade gliomas (pLGGs), defined as World Health Organization (WHO) grade I or II malignancies encompassing a wide array of histologies, the pilocytic histological subtype is commonly diagnosed in the first five years of life, while the fibrillary subtype is found at age of 10, accounting for 75% and 7% of all intrinsic pediatric spinal cancer, respectively [3,8]. High-grade gliomas (HGGs), defined from the WHO as grade III and IV, are very uncommon. Pilocytic astrocytoma is usually located at the cervicomedullary or cervicothoracic junction, and shows (i) “expansive” rather than “infiltrative” features (Figure 1), with (ii) spinal cord parenchymal “epicenter”, (iii) eccentric growth, and (iv) well-demarcated margins [3,7]. The swelling spine appearance could extend through several vertebral segments (usually < 4) [7], up to being holocordal—in this case it is important (but often complicated and not feasible) to distinguish between tumor and neoplastic edema [3]. The neoplasm could be predominantly (i) solid (40% of cases), (ii) necrotic cystic (60% of cases), or (iii) nodular cystic (Figure 2). Therefore, it could appear (i) iso/hypointense in T1 and hyperintense in T2 or (ii) hyperintense in T1 and T2, respectively [3]. The contrast enhancement is absent in 30% of cases [9] and, when present, is variable and depends on the different components. However, it is less evident than that of ependymomas [7]. Outcome of pLGGs depends on the extention of surgical resection and the success of the therapies—sometimes it is difficult to have a complete total resection. Near total resection is correlated with a 5-year progression free survival of up to 80% and overall survival of up to 95% [7].

Infiltrative growth, non-demarcated margins, and dissemination through the cerebrospinal fluid (CSF) are more typical in high-grade tumors. pHGGs account for about 0.2–1.5% [7] of all spinal astrocytomas and are more common in the cervical (Figure 3) and thoracic tract, while localization in the medullary cone is more rare (3% of pediatric spinal glioblastomas) [10,11]. The neoplasm usually shows inhomogeneous signal in T1 and T2 sequences, inhomogeneous enhancement, and also bleeding and cysts [10] (Figure 4 and Figure 5). The pHGG’s prognosis remains bleak, with a mean survival of 12 months after diagnosis and a range between 6–16 months [12].

The genetic landscape of pediatric spinal tumors has been less studied than that of intracranial localization: the involvement of KIAA1549–BRAF (v-raf murine sarcoma viral oncogene homolog B1), BRAFV600E, PTPN11, H3F3A, TP53, FGFR1, and CDKN2A deletion has been detected in pediatric spinal tumors [13,14]. One of the most studied molecular alterations concerns LGG and it is KIAA1549-BRAF fusion that causes the hyperactivation of the MAPK/ERK pathway.

It has been largely investigated and demonstrated that KIAA1549-BRAF-fused cerebellar pLGGs have a better prognosis compared to the ones not carrying the fusion; the same evidence has been described for the spinal low-grade gliomas [13,15]. On the other hand, the presence of BRAF fusion in the tumor suggests the potential use of target therapy, such as MEK inhibitors [16], while the use of BRAF inhibitors are indicated and useful when the BRAFV600 mutation is present [17]. However, the association between KIAA1549-BRAF fusion and outcome has not yet been validated, as according to some studies, this molecular alteration could not predict the prognosis [18]. Mutations in genes encoding histone H3.1 (HIST1H3B) and H3.3 (H3F3A) have been considered as a hallmark of diffuse midline gliomas. Instead, histonic mutations are also rarely found in pediatric low-grade gliomas [19]; this concept is important in highlighting the pathological heterogeneity in gliomas and the overlapped genetic landscape between high- and low-grade gliomas [13,20].

In the landscape of spinal pediatric LGG, clinical outcome can be related to type of resection, age, and presence of metastasis: there is a known link between subtotal resection, young age, and metastatic spreading at diagnosis with less event-free survival (EFS). A primary management goal for pediatric spinal tumors is to extend long-term follow-up because progression is possible even 10–20 years after diagnosis; for this reason, it is important to expand our knowledge around these tumors [18].

Ganglioglioma, most common in the first five years of life, accounts for about 15% of all intramedullary pediatric tumors [8]. These neoplasms, usually low-grade, may be characterized by local recurrence, but very uncommonly by the risk of malignant evolution (usually when both a BRAFV600E and a TP53 or CDNK2A/B deletion are present) [21,22]. The cervical and thoracic tracts are more frequently affected, with holocord involvement, related to slow growth [23]. The presence of calcifications is closely related to the diagnosis of ganglioglioma [8], while solid components and cysts [1], as well as edema and enhancement, are non-specific [24,25]. Due to relative rarity, few studies have focused on the molecular alterations of spinal gangliogliomas. The most common genetic modification is the V600E mutation in the BRAF oncogene [26].

### 2.2. Diffuse Leptomeningeal Glioneuronal Tumor

DL-GNT is a rare neuronal–glial tumor that was first defined in 2016. It is more common in males, with a median age at diagnosis of 4–6 years [6], characterized by slow growth and leptomeningeal dissemination. MRI may show different features, such as multiple spinal and/or intracranial (i) leptomeningeal nodular lesions with diffuse sheet enhancement (the most frequent findings), (ii) cystic-like leptomeningeal nodular lesions (T2 hyperintensity), (iii) intraparenchymal lesions [27,28], (iv) an intraparenchymal spinal/intracranial mass, and (v) nonenhancing cystic-like small intra-axial lesions (Figure 6) [27]. Differential diagnoses include infections, leptomeningeal dissemination of other neoplasms, lymphoproliferative diseases, and neurosarcoidosis (rare in pediatric age).

The typical molecular landscape of DL-GNT is characterized by alterations leading to an aberrant MAPK/ERK pathway, together with loss of chromosomal arm 1p (sometimes with 19q co-deletion) [29]. The alteration of the MAPK/ERK pathway is most commonly caused by fusion of KIAA1549-BRAF, but also NTRK1/2/3 and TRIM33:RAF1. Some authors have hypothesized a classification in two molecular subgroups, which include a DLGNT methylation class (MC)-1, characterized by lower median age (5 years), with a more favorable prognosis. DLGNT (MC)-2 is identified by loss of chromosomal arm 1p together with gain of chromosomal arm 1q, with median age of 14 years and a worse prognosis [29]. Survival in this subtype of pLGG is highly variable because the tumor shows slow but disseminated growth and can cause secondary hydrocephalus, which increases morbidity [30].

## 3. Ependymomas

Ependymomas account for 30% of pediatric intrinsic spinal cancers [4] and are closely related to a specific type of phacomatosis (genetic neurocutaneous disorders), that is type 2 neurofibromatosis (NF2), which can be in multiples. Usually, cervical ependymomas [4] are characterized by (i) a central spinal epicenter, (ii) clear delimitation, and (iii) expansive (and slow) growth rather than infiltrative [3,7]. MRI commonly reveals a T2 low-signal rim at the caudal or cranial margins (the typical “cap sign” related to internal bleeding), vivid enhancement with marked vascularization [31], and polar cysts rather than intratumoral [4] (Figure 7). Nine distinct molecular subgroups of ependymomas have been identified, three within each anatomical compartment within the CNS: supratentorial, posterior fossa, and spinal cord. Within the spinal cord, the three distinct subgroups correspond to the WHO histologic subtypes: myxopapillary ependymoma, subependymoma, and classic ependymoma. Loss of chromosome 22q (NF2 locus) is a frequent finding; NF2 is a tumor suppressor gene located at 22q12.2 and is the only known driver of spinal ependymoma (30). Although rare, N-Myc amplification can occur in anaplastic ependymoma of the spinal cord and is frequently associated with aggressive clinical behavior. CSF spreading is possible, but is more frequent within this N-Myc subtype [32,33] and is associated with poor prognosis among spinal ependymomas.

An intradural-extramedullary neoplasm, myxopapillary ependymoma (Figure 8), is typically located in the lumbo-sacral region due to its origin in the filum terminale [34]. This subtype represents 13% of all spinal ependymomas and can also extend into the neuroforamina, thus differential diagnosis includes extradural tumors [3].

Outcomes in pediatric spinal ependymomas are related to the presence of dissemination, N-Myc mutation, and surgery: the 5-year survival is about 80% in cases of gross total resection and 57% in cases of non-gross total resection [7,35].

## 4. Mesenchymal, Non-Meningothelial Tumors

### 4.1. Hemangioblastomas

As low-grade tumors with variable epicenters, hemangioblastomas are uncommon in children. Neoplastic growth is slow and can be intramedullary (75% of cases), but also intradural when tumors arise from nerve roots or extradural [1]. MRI usually shows a tumor with (i) predominantly solid component and clear margins, (ii) rich vascular support, and (iii) avid enhancement and sometimes cysts (Figure 9) and bleeding [7]. Before surgical excision, it is useful to perform a medullary angiography to identify arterial feeders that can be embolized. Hemangioblastomas are commonly found in patients with von Hippel-Lindau (VHL) syndrome and findings of hemangioblastoma should prompt investigation for VHL. Hemangioblastomas are formed by “stromal” cells (with VHL mutation) and rich blood vessels (without VHL mutation). This evident angiogenesis is related to the activation of angiogenetic factors (VEGF, HIF) in the stromal cells [36]. Surgical excision is the treatment of choice and can be associated with a preventive embolization [7].

### 4.2. Mesenchymal Chondrosarcomas

Mesenchymal chondrosarcomas (MSCs) are very rare pediatric intraspinal tumors. Classical chondrosarcoma most frequently affects adults, while MCS affects children and young adults, accounting for 2–10% of all chondrosarcomas [37]. Usually with skeletal origin, MCS can also arise in other locations (25% of cases), such as the brain, meninges, and spinal cord [38], with a preference for the thoracic tract. This tumor is characterized by a potential aggressive course, metastasis, and poor prognosis, although the data on survival are highly variable due to the rarity and the different locations [39]. The genetic hallmark in pediatric extraskeletal MCS is the HEY1/NCOA2 fusion [40]. Due to the different site of origin, mesenchymal chondrosarcomas can have different imaging characteristics. Tumors may show inhomogoneous T2 hyperintensity and inhomogeneous enhancement, but these features are related to the specific case. For all these features, the outcomes of pediatric patients with mesenchymal chondrosarcomas are highly variable: some data report a survival of 88.9% at 5 years associated with radio/chemotherapy treatment [39,41].

## 5. Meningiomas and Tumors of the Paraspinal Nerves

Pediatric schwannomas develop from Schwann cells and are rarely sporadic. They account for 0.3% of intraspinal tumors [42] and are most commonly associated with NF2. When their localization is the spine, they may arise from the intra- (more common) or extradural tract of the spinal root with variable associated signs and symptoms, such as bone remodeling. The most important feature is that the lesion usually shows expansive (and non-infiltrative) growth relative to the nerve root, a well-defined capsule, and a plane of cleavage [3]. In the T1- and T2-weighted sequences, the tumor appears iso-/hypo-intense and iso-/hyper-intense, respectively, with homogeneous enhancement [3]. Differential diagnoses of spinal schwannomas are those of plexiform neurofibromas, which are intradural-extramedullary tumors. With similar signaling and enhancement features, neurofibromas differ from schwannomas in that they include both Schwann cells and fibroblasts, have infiltrative rather than expansive growth towards the spinal root [3], and could have a malignant evolution with rapid development [43]. Sometimes in neurofibromas, it is possible to observe the “target sign” (central area of hypointensity in T2) due to the high stromal component of collagen [43]. In pediatric patients with malignant peripheral nerve sheath tumors, the 5-year event-free survival (EFS) has been shown to be around 52.9% and overall survival (OS) is around 62.1% [44].

Meningiomas, rare in children, make up only 3% of pediatric SNC tumors [45] and are closely related to the diagnosis of NF2; it is estimated that 20% of NF2 patients harbor spinal meningiomas. Meningiomas in NF2 are typically WHO grade 1, slow-growing, benign tumors. When present, meningiomas in NF2 patients are often multiple, which contributes significantly to morbidity and mortality. The “clear cell meningioma (CCM)” (WHO grade II) is a typical pediatric/juvenile spinal meningioma and represents the most common histological subtype of sporadic pediatric spinal meningioma [3]. They are characterized by early local recurrence and cerebrospinal fluid metastasis. The World Health Organization defines CCM as a grade II cancer. Its incidence rate in children is higher than that in adults, who may manifest aggressive features such as recurrence and CSF dissemination [46,47,48,49]. The MRI features include dural tail, isointensity in T1 and T2 compared to the spinal cord, clear margins, and homogeneous enhancement (Figure 10) [3]. Outcome is closely related to surgery: gross total resection is associated with a good clinical outcome, while subtotal resection or partial resection could be associated with recurrence or growth of the residual lesion with clinical progression even 60 months after the first surgery (long-term recurrence) [50].

Some authors suggest a complete clinical, radiological, and genetic evaluation in single, apparently sporadic, cranial/spinal meningioma and schwannoma [42]. Beyond the NF2, the SMARCE1 (SWI/SNF-related, matrix-associated, actin-dependent regulator of chromatin, subfamily E, member 1) gene mutation was found in young adults (16–24 years) with meningioma. Heterozygous loss-of-function mutations in the SWI/SNF chromatin-remodeling complex subunit gene SMARCE1 play a key role in the pathogenesis of spinal meningiomas with clear cell histology [51]. Furthermore, in young people (1–15 years), the germline LZTR1-mutation is associated with the development of schwannomas, as well as that of SMARCB1, which, in the older age group, accounts for 25% of schwannomatosis disease [42].

## 6. Embryonal Tumors

Embryonal tumors with spinal localization are very rare. In our opinion, among these, it is useful to describe the atypical teratoid/rhabdoid tumors and embryonal tumors with multilayered rosettes, which are rarely found in children.

### 6.1. Atypical Teratoid/Rhabdoid Tumor

Atypical teratoid/rhabdoid tumor (ATRT) is a rare brain tumor that is more common in children under 2 years of age and accounts for 1–2% of all pediatric brain neoplasms [51,52,53] with aggressive development and poor prognosis [53,54]. Intramedullary spinal atypical teratoid/rhabdoid tumor (spATRT) origin accounts for 3.5% of all ATRTs, and intradural-extramedullary and extradural cases have also been described. MRIs show an inhomogeneous lesion and iso/hypointense in T1 and T2 due to the presence of solid and/or cystic components, necrosis, and bleeding with heterogeneous enhancement [3,52]. There are three molecular subgroups based on DNA methylation, which include ATRT-MYC, ATRT-SHH, and ATRT-TYR [55,56], but currently there is no recognized correlation with localization, response to treatment, and prognosis due to the few cases of spATRT [57]. Moreover, the deletion or mutation of the SMARCB1 locus, a typical genetic alteration, was detected [52,58,59]. Outcome of children with ATRT is poor and the median survival is about 17 months [60,61].

### 6.2. Embryonal Tumor with Multilayered Rosettes

Embryonal tumor with multilayered rosettes (ETMR), C19MC-altered, is a rare and aggressive, high-grade neoplasm, more common in infants (<3 years old) [62,63], with an average survival of about 1 year [64]. Introduced in 2016, ETMR encompasses previous diagnostic entities, such as embryonal tumor with abundant neuropil and true rosettes (ETANTR) [65], ependymoblastoma (EBL), and medulloepithelioma (MEPL) [6,66]. ETMRs have frequent brain localization (70% supratentorial; 30% infratentorial) and are exceptionally pronounced in the spinal cord (<1%) [67]. MRIs show a large tumor with restricted diffusion, cystic components, and heterogeneous pre-contrast signal and enhancement [68,69]. The genetic hallmark of this neoplasm is the C19MC amplification (90% of all ETMRs), and it is possible to identify (in the rosette-forming cells) LIN28A-positive immunostaining (RNA-binding protein) [69]. Half of the ETMR cases that do not show C19MC amplification are characterized by biallelic DICER1 inactivation [67].

## 7. Genetic Syndromes

Some genetic syndromes are associated with the risk of developing central nervous system (CNS) and extra-CNS tumors. They include: ataxia telangiectasia, Cowden syndrome, familial adenomatous polyposis, hereditary non–polyposis-related colorectal cancer, Li-Fraumeni syndrome, Gorlin syndrome, multiple endocrine neoplasia type 1, tuberous sclerosis complex, Turcot syndrome [70], and DICER1 syndrome [67].

In this context, there are few familial syndromes closely associated to the development of spinal tumors. Gliomas may be related to neurofibromatosis type 1 (NF1). This genetic syndrome has an incidence rate of 1 in 3000 people [71], caused by a germline mutation in the NF1 gene. The diagnostic criteria for NF1 [72] are met in an individual who does not have a parent diagnosed with NF1 if two or more of the following are present: six or more café-au-lait macules over 5 mm in greatest diameter in prepubertal individuals and over 15 mm in greatest diameter in postpubertal individuals, freckling in the axillary or inguinal regiona, two or more neurofibromas of any type or one plexiform neurofibroma, optic pathway glioma, two or more iris Lisch nodules or two or more choroidal abnormalities, a distinctive osseous lesion such as sphenoid dysplasia, anterolateral bowing of the tibia, or pseudarthrosis of a long bone. NF1-gliomas are most commonly optic/chiasmatic/hypothalamic pilocytic astrocytomas; on the other hand, although uncommon, they can be diffusely infiltrating astrocytomas [73] and originate in the spinal cord [74]. The most common spinal tumors in NF1 remain paraspinal neoplasms (and in this case they are overall plexiform neurofibromas) (Figure 11), while intramedullary tumors occur only in 2–6% of patients with NF1 [75]. “Spinal-NF1” refers to a particular subgroup of patients who show multiple paraspinal tumors (and few skin lesions) associated with a large deletion on the NF1 gene [76]. Malignant peripheral nerve sheath tumors occur in 2–10% of patients with NF1 and are very rare in the general population (0.001%) [77]. New neurological symptoms/deficits, pain, changes in the growth and consistency, compression of the locoregional areas, and bleeding of the neurofibroma are the so-called “red flag” symptoms [78] suspected for aggressive evolution of the neurofibroma (10% of cases [79]) or aggressive paraspinal neoplasm. Primary (related to dysplasia) and secondary (related to the tumor growth) bony remodeling, scoliosis, dural ectasia, and lateral meningocele are other spinal manifestations of NF1 [78]. In addition to the supra and subtentorials, spinal UBOs (unidentified bright objects) can be observed.

On the other hand, the diagnosis of spinal ependymoma can be linked to neurofibromatosis type 2 (NF2, 1 in 60,000 people [71,80], germ-line mutation in the NF2 gene). Family heredity is recognized in half of the patients with NF2, while in the remaining cases, a “de novo” mutation is hypothesized. NF2 may show schwannoma (typically bilateral vestibular tumors) and cranial/spinal meningiomas and ependymomas [81]. Spinal meningiomas are seen in approximately 20% of patients with NF2 [82,83] and often belong to the fibrous variant [83,84]. In 50% of the patients, the NF2 manifestations occur from the second decade of life; however, related to the type of genetic alteration, two clinical subtypes are recognized. The Wishart phenotype (Figure 12) is associated with more severe diseases, tumors onset before the age of 20, rapid progression, and truncating alterations in NF2 gene, while the Gardner phenotype shows milder disease, later and slower onset of tumors, and missense loss-of-function mutations in the NF2 gene [83,85].

The diagnosis of spinal, retinal, or cerebellar hemangioblastoma may be closely related to the von Hippel-Lindau (VHL) tumor syndrome (1 in 36,000 people [86,87]; germline mutation in VHL gene). VHL is a cancer predisposition syndrome associated with benign and malignant SNC/extra-SNC neoplasms. The most common tumors are hemangioblastomas, endolymphatic sac tumors, pheochromocytomas, paragangliomas, renal tumors, and cystic and pancreatic neuroendocrine tumors. Up to one-third of patients (10–30%) with VHL develop spinal cord hemangioblastoma [36,59] and the symptoms may be due to bleeding.

To conclude, spinal tumors, especially in children, can be an indicator of a cancer predisposition syndrome, such as NF1, NF2, and VHL syndrome. For this reason, in the diagnostic setting of a pediatric spinal tumor, it is essential to perform an MRI of the entire neuroaxis and a familial genetic evaluation. 

## 8. Imaging Technique and Differential Diagnoses

Cancer predisposition syndromes (CPSs) differ from each other in relation to the site of onset of the neoplasm, type of neoplasm, and involvement (or not) of the CNS. In this setting, whole-body MRI is the preferred imaging modality for surveillance of pediatric patients with CPSs, and the growing literature supports its importance in presymptomatic cancer detection, but further studies are needed and the question is still open [88]. Nevertheless, evaluation and follow-up of children with CNS tumor, not only in the CPSs, is based on brain and spine MRI.

MRI is the standard reference for the evaluation of spinal tumors, surgical planning, and surveillance during and after treatment. Entire neuroaxis imaging using contrast-enhanced MRI should be performed on all patients with a spinal tumor to detect other disease sites in addition to the primary spinal lesion (spreading metastases starting from the spinal cancer) or an intracranial primary neoplasm. In the latter case, spinal lesion/s can represent spinal disease dissemination starting from the brain, such as drop metastases.

Currently, MRI protocol includes a standard pre-contrast study, such as sagittal (and axial) T1 and T2 sequences (thickness: 3 mm) and axial, sagittal (and coronal) T1 sequences after contrast [89]. In addition, heavily T2-weighted MR “cisternography” sequences (e.g., CISS, “constructive interference in steady-state” and DRIVE, “3D driven equilibrium”), fat suppression sequences (T1- or T2-weighted), diffusion-weighted imaging (DWI), 3D volumetric sequences, and susceptibility-weighted imaging (SWI) can be used. The 3D-CISS sequence, characterized by a high spatial resolution, allows a detailed view (Figure 13 and Figure 14) of small components (for example spinal roots, cysts) with three-plane visualization (isotropic sequence) [90]. SWI, related to the magnetic susceptibility of the different tissue components, has a recognized role in the brain but its role is limited and not standardized in the spinal cord [90]. In the brain, it is used to detect bleeding, calcifications, iron, and deoxygenated hemoglobin; in the spine, its potential uses include the evaluation of venous anatomy [91] and, therefore, the results of vascular malformation treatments [92] and bleeding (potential use in ependymoma). Advanced neuroimaging methods, such as diffusion-weighted imaging (DWI), can be useful techniques, mostly to increase conspicuity in the detection of spinal metastases. The DWI technique is based on the motion of extracellular water molecules—the increase in the number of cells (neoplasms with high cellularity) causes a reduction in the movement of extracellular water (decreased diffusion of extracellular water; high cellularity = diffusion restricted) [93].

Advanced techniques not yet standardized in clinical practice and protocols include diffusion tensor imaging (DTI), perfusion technique, spectroscopy, and functional imaging. Movement artefacts, due to the heart and lungs and the reduced size of the spinal cord (compared to the cerebral hemispheres), limits the use of DTI and DWI. DTI can be useful in the differential diagnosis between astrocytomas and ependymomas: in the first case, MRI demonstrates involvement of the spinal cord fibers within the tumor, and in the second, case shows dislocation of the fibers [94]. On the other hand, the relationship between tumor and spinal cord fibers can help surgical planning. (i) Acute ischemic events and (ii) the differential diagnosis between infectious and chronic inflammatory degenerative disease are the main situations in which DWI plays a primary role [95].

In the spinal cord, a particular use of the dynamic contrast enhancement (DCE), the most common perfusion technique, is to study extradural metastases, their vascularization, and the possibility of endovascular treatment [96]. Although spectroscopy is rarely included in a spinal cord protocol study, it may be useful in the differential diagnosis between neoplasm [97] and inflammation, in amyotrophic lateral sclerosis, and in predicting neurological outcomes for patients affected by cervical spondylotic myelopathy after surgical treatment (the latter two in adult patients) [90,98]. Functional imaging (fMRI) is currently for research purposes only. 

Spinal tumors disrupt the balance between the spinal cord, cerebrospinal fluid (CSF), and the spinal canal: intrinsic neoplasms cause expansion of the spinal cord, obstacles to CSF flow, and, sometimes, bone remodeling, with associated signs and symptoms. However, back pain and sensory and motor symptoms in a child are nonspecific symptoms and must be investigated. The main differential diagnoses include infectious, inflammatory, demyelinating, vascular diseases, “tumor-like” lesion, and “tumor-like” mass, in particular when there is a spine signal alteration without swollen appearance (Figure 15, Figure 16 and Figure 17). When an MRI is performed, it can show a single (or multiple) focal or diffuse area of altered medullary signal, usually T2 hyperintense, with or without (i) swollen appearance (tumor-like appearance) of the spinal cord and (ii) enhancement—in these cases, acute transverse myelopathy (ATM) should be considered between the differential diagnoses [99]. ATM represents a set of diseases and includes idiopathic (idiopathic ATM or acute transverse myelitis), compressive, post-infectious (subacute disseminated encephalomyelitis, ADEM), viral, demyelinating (multiple sclerosis), ischemic, and autoimmune forms. It is very useful to investigate for diagnosis: (i) previous traumatic events, to exclude spinal cord mechanical compression; (ii) the time to onset, which is very short in spinal ischemic vascular disease; (iii) recent medical history, which may include infection or vaccination one or two weeks prior to onset symptoms in ADEM; and (iv) associated symptoms, such as visual disturbances in multiple sclerosis [99]. Moreover, an MRI may show specific findings, such as a typical vascular pattern in spinal ischemia and involvement of the white matter of the brain and spinal cord in ADEM (widespread T2 hyperintensity) and multiple sclerosis (usually multiple focal T2 hyperintensities) [99]. However, the idiopathic form is a diagnosis of exclusion. 

In the context of an infectious inflammatory pathology, meningitis can have a “tumor-like appearance” with particular radiological pictures. More frequently, bacterial infection of the meningeal sheets is due to different age-related pathogens such as Streptococcus B (more common in newborns), Haemophilus influenzae (young infants), pneumococcus, N. Meningitidis, staphylococci (in older children), and, rarely, Mycobacterium tuberculosis. Diagnosis is based on clinical and laboratory criteria, and imaging is useful in selected cases when there is suspicion of complications. On MRI, the involvement of the meninges causes enhancement of the pachi- and/or lepto-meninges, the surface of the spinal cord, and the nerve roots [99]. Moreover, in intracranial tuberculosis, MRI can demonstrate signs of meningitis but also tuberculomas and abscesses [100]. These findings represent infectious leptomeningeal dissemination and, in some cases, the differential diagnosis from tumor leptomeningeal involvement may be difficult based on imaging alone, especially when there is a history of aggressive tumor of the CNS. Clinical condition, laboratory tests, lumbar puncture, and response to antibiotic or antiviral therapy clarify the infectious origin of the disease.

Finally, in the landscape of a patient with NF1, spinal UBOs are sometimes underestimated. Asymptomatic UBOs are usually T2 hyperintense intramedullary areas (Figure 18), without evident enhancement and sometimes with “tumor-like appearance”. Due to these findings, UBOs may initially be diagnosed as low-grade glial spinal neoplasms [101]. The definitive diagnosis of UBOs is related to (i) the absence of “spinal” symptoms, (ii) the clinical stability, and (iii) the progressive spontaneous regression to surveillance MRI [101].

## 9. Targeted Therapies

Surgical resection is the mainstay of treatment for those spinal tumors not suitable for only observational follow up—surgery is undertaken when feasible and safe to perform with curative intent, while debulking procedures are reserved most frequently for symptomatic relief. The recent WHO classification for brain tumors considers data from genomic sequencing studies to incorporate molecular characteristics helpful for targeted therapies [102]. Due to the rare incidence of spinal cord neoplasms and to the heterogeneous histologies, their molecular features remain unclear and still poorly known.

When considering systemic therapies, as frontline, but more often as adjuvant treatment, in sporadic pLGGs, genomic alterations in MAPK pathways are the most common molecular characteristics.

The presence of BRAF fusion KIAA1549:BRAF suggests the potential use of target therapy such as MEK inhibitors [16], as well as the use of BRAF inhibitors when its mutations are present [17,103].

The most common point mutation in PAs is the BRAFV600E mutation, identified in 17% of pLGGs. Clinical trials of the first generation BRAF inhibitor dabrafenib reported up to 44% response rate in pLGGs [104].

Limited information is available about NTRK alterations in cerebral pLGGs and only speculative interest relates to spinal pLGG. RTK inhibitors are efficacious in several pathways, so more information is needed [105].

Phosphoinositide 3-kinase/protein kinase B/mTOR (PI3K/Akt/mTOR) pathway mutations are very common and everolimus has been explored as treatment when this pathway is overexpressed [106].

Although IDH1 mutations frequently occur in brain astrocytomas and rarely in pediatric age, the incidence of IDH1 mutations in spinal cord astrocytomas has been found to be rather low, excluding enasidenib and ivosidenib from potential therapies [107].

Even though it is difficult to distinguish the response to therapy from the natural history of the tumor, there are numerous target drugs that have been studied and are potentially useful in VHL syndrome: monoclonal antibodies (bevacizumab, ranibizumab, pegaptanib), tyrosine kinase inhibitors (semaxanib, sunitinib, pazobanib, erlotinib, dovitinib, sorafenib), and biological response modifiers [36].

Given the evidence that the SH3PXD2A-HTRA1 fusion is a potential driver present in a subset of schwannomas, Agnihotri et al., in preclinical data, provided a rationale that fusion-positive cells are potentially sensitive to MEK inhibitors and may represent a therapeutic approach for treatment-refractory fusion-positive schwannomas [108].

## 10. Conclusions

In conclusion, pediatric spinal tumors are less frequent than brain neoplasms and should be suspected when children have “nonspecific” sensory and motor symptoms. 

MRI is the reference imaging method, but clinical history is important. When MRIs show an “atypical” focal or diffuse area of altered intra/extramedullary signal, usually T2 hyperintense with (or without) swollen appearance (tumor-like appearance), it is important to know the onset (acute or chronic) of symptoms, previous traumatic events, recent infections, and the results of laboratory tests. These findings are essential to investigate differential diagnoses, as well as plan therapeutic strategies that may include surgical excision or biopsy and adjuvant therapies when needed. 

An MRI evaluation includes pre- and post-contrast T1-weighted sequences, T2-weighted sequences, and MR “cisternography” sequences; DWI plays an important role in the assessment and detection of metastases. The role of advanced neuroimaging (DTI, DCE, spectroscopy, and functional imaging) is not clearly standardized in the study of spinal tumors and these methods are not yet included in a “standard” spinal cord evaluation. 

Currently, it is essential to investigate the potential molecular alterations associated with pediatric spinal cancer that provide novel and additional information regarding the tumor, in order to improve both diagnosis and therapeutic strategies that include (new) target drugs (Table 1, Figure 19, Figure 20 and Figure 21). Moreover, the diagnosis of spinal tumors, especially in children, can be an indicator of a cancer predisposition syndrome such as NF1, NF2, and VHL syndrome. Therefore, when a pediatric spinal tumor is found, it is necessary to perform an MRI of the entire neuroaxis and a familial genetic evaluation.

## Figures and Tables

**Figure 1 diagnostics-11-01710-f001:**
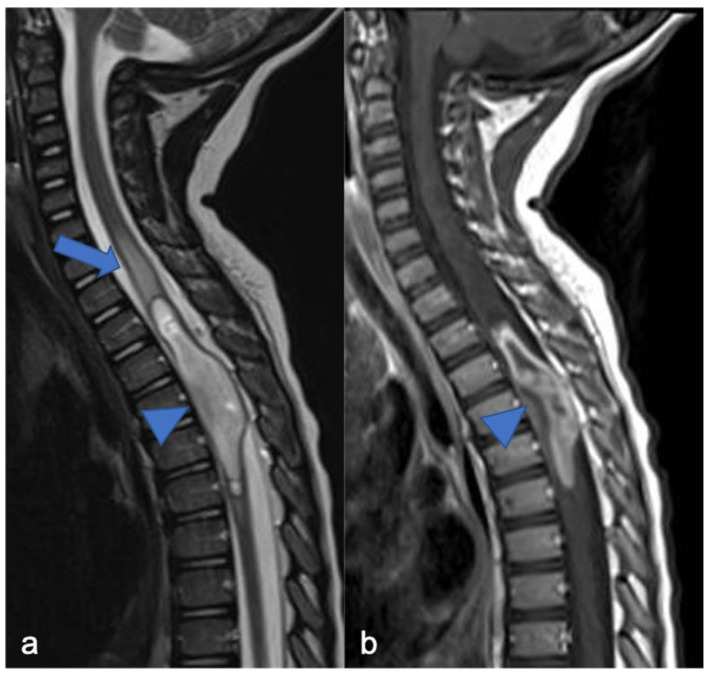
Pilocytic astrocytoma in a two-year-old child. Sagittal T2-weighted (**a**) and post-contrast T1-weighted (**b**) images show intramedullary hyperintense mass with inhomogeneous contrast enhancement, respectively (arrowheads). There is perilesional spinal cord edema (arrow).

**Figure 2 diagnostics-11-01710-f002:**
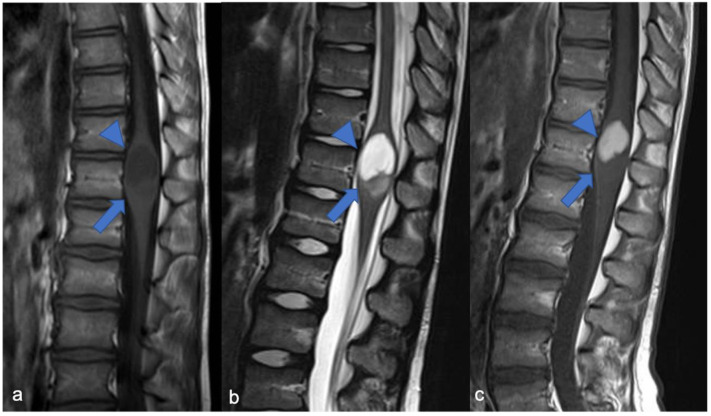
Pilocytic astrocytoma in an eight-year-old child, including expansive mass and distinct solid cystic components at level D11-12. Sagittal T1-weighted (**a**), T2-weighted (**b**), and post-contrast T1-weighted (**c**) images demonstrate a cystic-like cranial component with evident and homogeneous enhancement (arrowheads) and a caudal solid component without enhancement (arrows).

**Figure 3 diagnostics-11-01710-f003:**
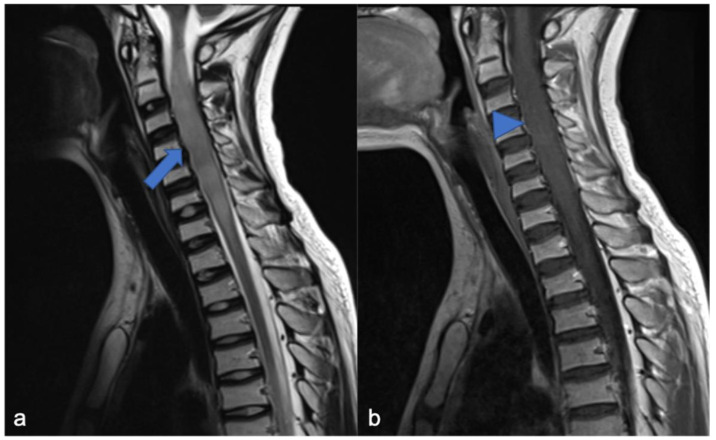
High-grade glioma with H3K27M mutation in a thirteen-year-old child. Sagittal T2-weighted (**a**) and post-contrast T1-weighted (**b**) images demonstrate swollen appearance of the cervical cord characterized by T2 mild hyperintensity (arrow) and poor enhancement (arrowhead).

**Figure 4 diagnostics-11-01710-f004:**
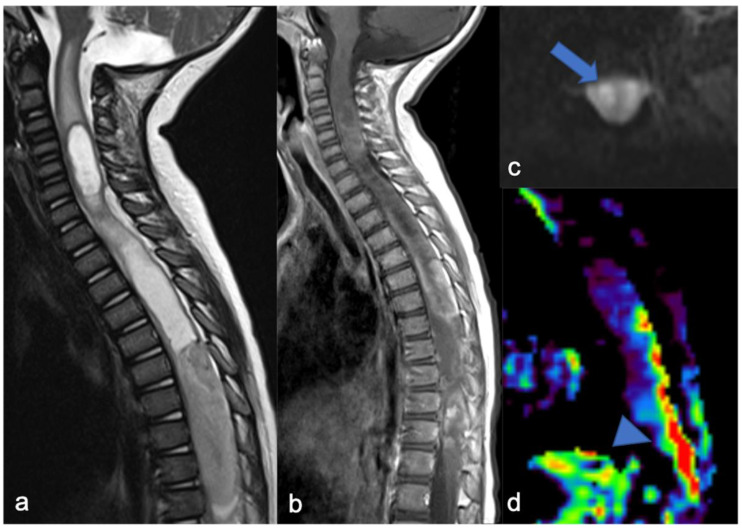
Sagittal T2-weighted image (**a**), post-contrast T1-weighted (**b**), DWI (**c**; T10-T11) and DSC (**d**). High-grade glioma with cervical–thoracic epicenter and holocordal involvement of the spinal cord in a two-year-old child. The neoplasm is characterized by inhomogeneous enhancement. Components with restricted diffusion (arrow) and increased rCBV (arrowhead) are shown.

**Figure 5 diagnostics-11-01710-f005:**
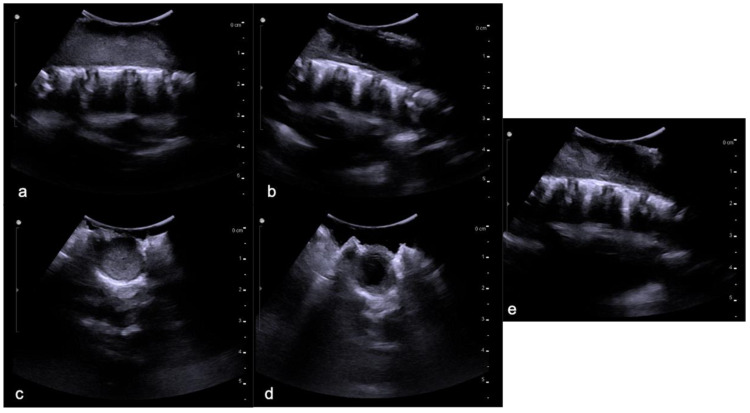
The same patient as in Figure 4 with high-grade glioma. Intraoperative ultrasound (transversal and longitudinal planes) shows the dorsal solid (**a**,**c**) and cystic-like (**b**,**d**) components and the cystic/solid transition zone (**e**).

**Figure 6 diagnostics-11-01710-f006:**
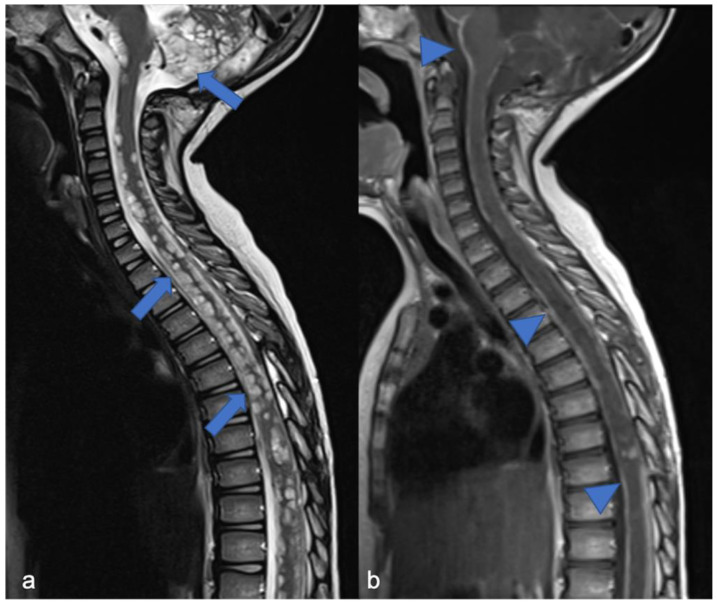
Diffuse leptomeningeal glioneuronal tumor in a four-year-old child. Sagittal T2-weighted (**a**) and post-contrast T1-weighted (**b**) images show multiple spinal and subtentorial cystic-like leptomeningeal nodular lesions (arrows) with diffuse sheets enhancement (arrowheads).

**Figure 7 diagnostics-11-01710-f007:**
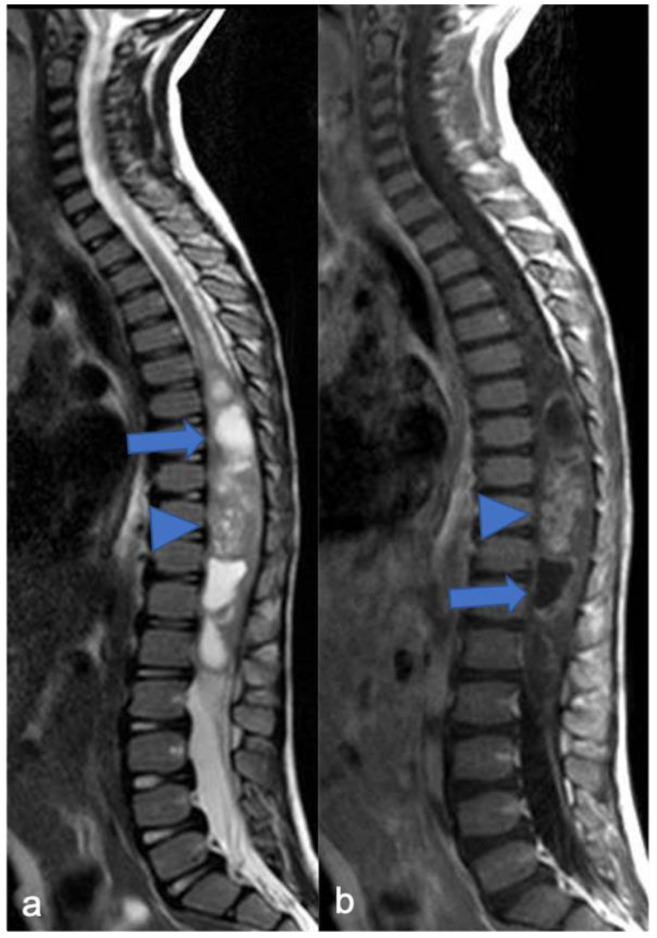
Ependimoma in a six-year-old child. Sagittal T2-weighted (**a**) and post-contrast T1-weighted (**b**) images show expansive lesion of the dorsal spinal cord with an enhancing solid nodular component (arrowheads) and a cystic-like «polar» component (arrow). In this case the “cap sign” is not clearly evident.

**Figure 8 diagnostics-11-01710-f008:**
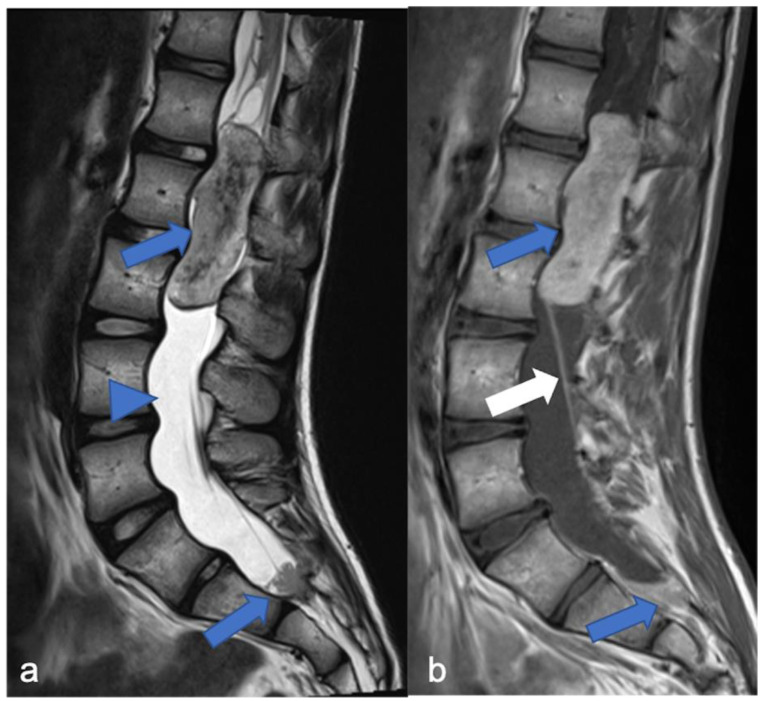
Myxopapillary ependymoma in a fourteen-year-old child located in the lumbo-sacral region. Sagittal T2-weighted (**a**) and post-contrast T1-weighted (**b**) images demonstrate solid cranial and caudal enhancing components (blue arrows) and a pseudocystic non-enhancing component (arrowhead). The detail of the filum terminale is also highlighted (white arrow).

**Figure 9 diagnostics-11-01710-f009:**
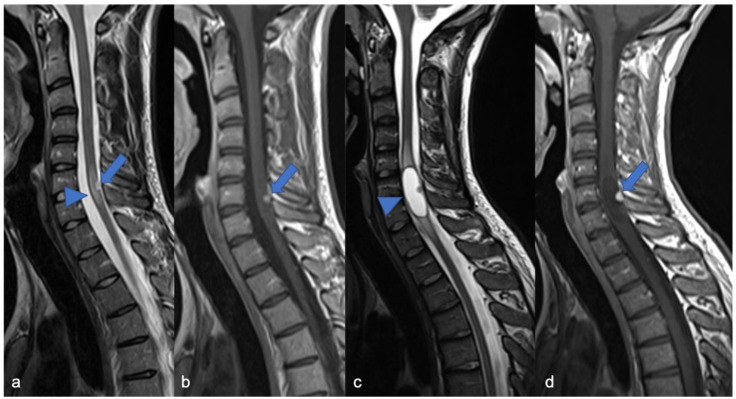
Hemangioblastomas in a fifteen-year-old child with cystic solid components located in the cervical–dorsal junction. Sagittal T2-weighted (**a**) and post-contrast T1-weighted (**b**) images demonstrate the enhancing solid nodular components (arrows) and the cystic component (arrowhead). After two years (**c**,**d**), the cystic component increased while the solid component remained stable.

**Figure 10 diagnostics-11-01710-f010:**
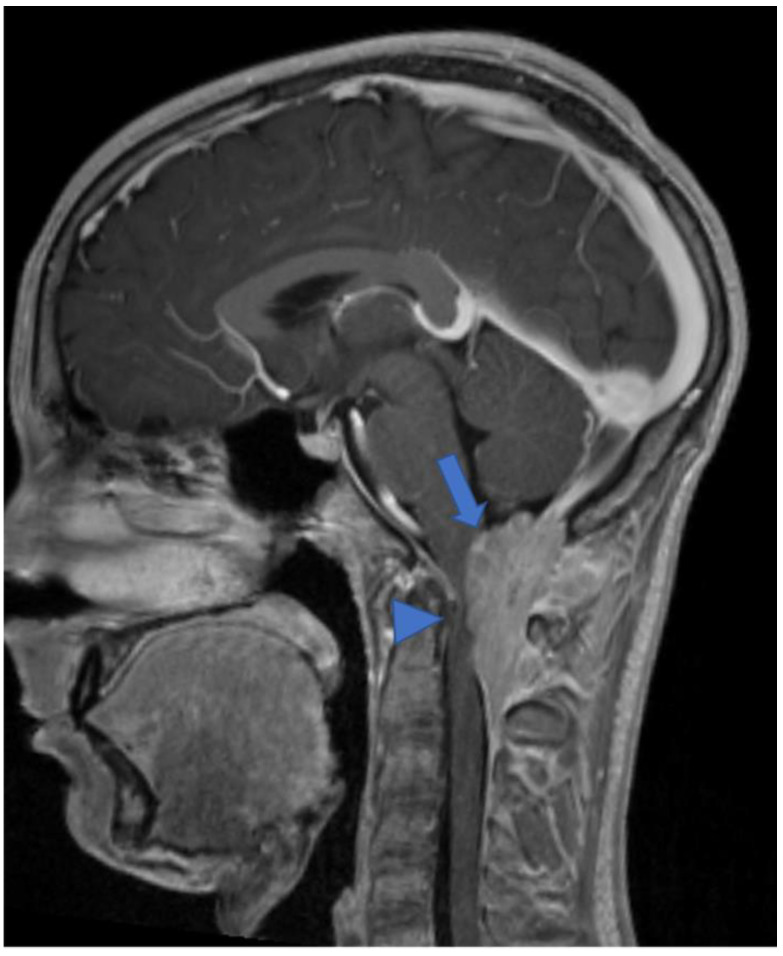
Gadolinium-enhanced T1-weighted image. Craniocervical junction meningioma (arrow) characterized by dural base, well-defined margins, and intense and homogeneous enhancement. Compression and anterior dislocation of the bulb and cervical cord are evident (arrowhead).

**Figure 11 diagnostics-11-01710-f011:**
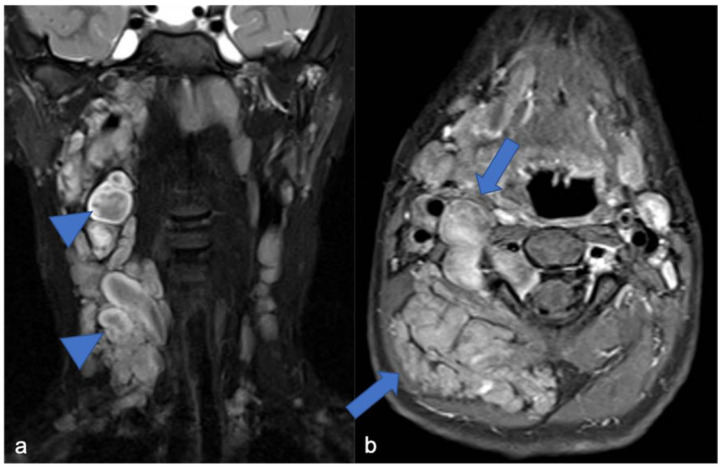
(**a**) Coronal «saturated» T2-weighted and (**b**) axial post-contrast «saturated» T1-weighted images. Multiple neurofibromas (arrows) in a pediatric patient with NF1. The “target sign” (central area of hypointensity in T2) can be observed (arrowheads).

**Figure 12 diagnostics-11-01710-f012:**
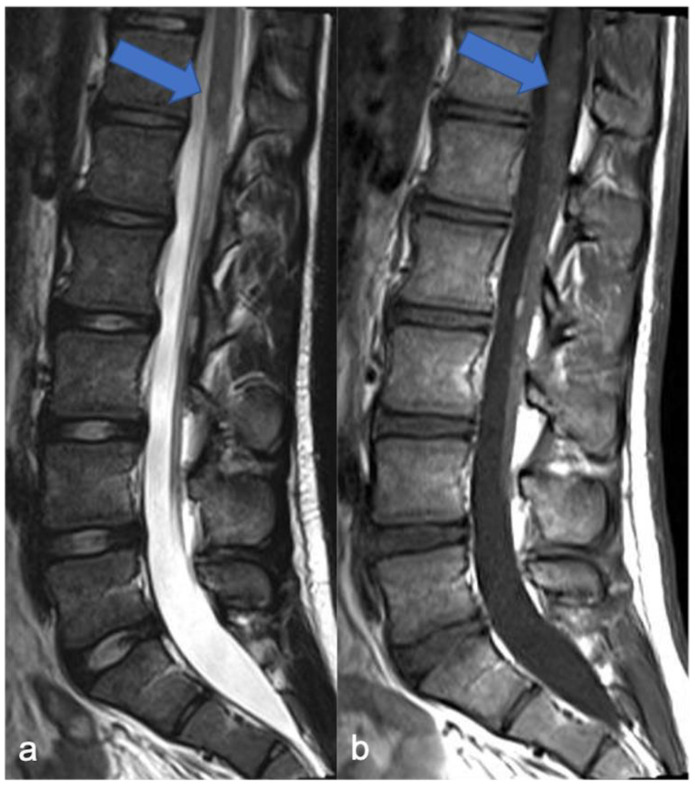
A detail of the lumbo-sacral spine in a twelve-year-old child with NF2 and Wishart phenotype. Sagittal T2-weighted (**a**) and post-contrast T1-weighted (**b**) images show an expansive intramedullary lesion of the cauda, hyperintense in T2 (arrows) with enhancement. These findings are compatible with ependymoma in patient with NF2.

**Figure 13 diagnostics-11-01710-f013:**
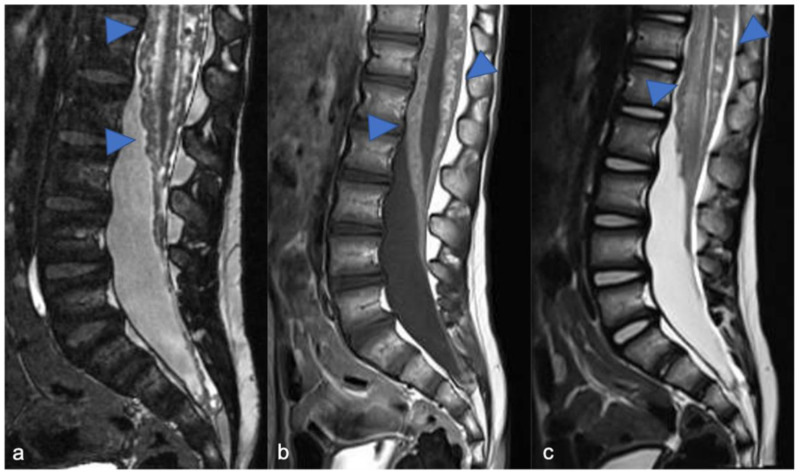
Leptomeningeal dissemination related to medulloblastoma. CISS (**a**), post-contrast T1-weighted (**b**) and T2-weighted (**c**) images show diffuse nodular appearance of the meningeal sheets (arrowheads). The panel demonstrates more detail of the CISS (**a**) than the T2-weighted image (**c**).

**Figure 14 diagnostics-11-01710-f014:**
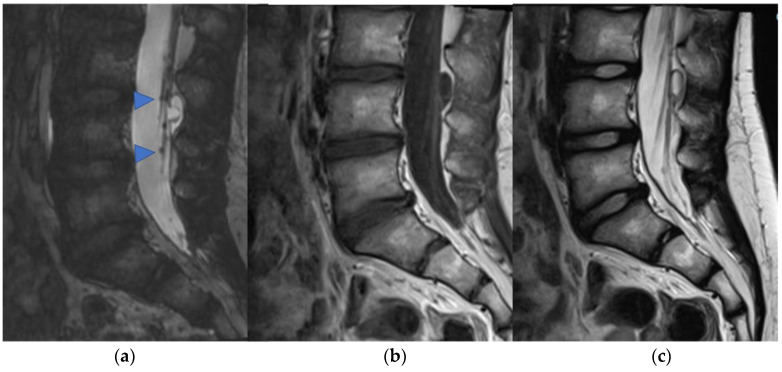
Another case of leptomeningeal dissemination related to medulloblastoma, in which the usefulness of the CISS (**a**) is clear compared to post-contrast T1-weighted (**b**) and T2-weighted (**c**) images. Nodular appearance of the cauda roots is evident (arrowheads).

**Figure 15 diagnostics-11-01710-f015:**
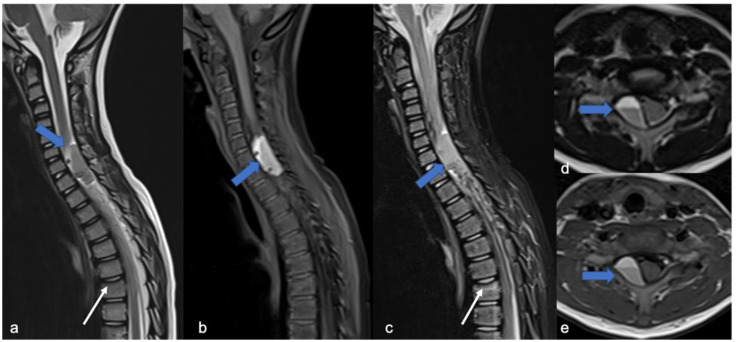
Tumor-like appearance of a spinal “formation” in a six-year-old child. MRI was performed after a traumatic event. Sagittal T2-weighted (**a**), fat suppression T1-weighted (**b**), fat suppression T2-weighted (**c**), axial T2-weighted (**d**), and T1-weighted (**e**) images demonstrate intradural-extramedullary cystic-like formation with a fluid–fluid level (blue arrows) and spinal cord dislocation. Traumatic deformation of the superior vertebral plateau is also evident (white arrows). The definitive diagnosis was post-traumatic pseudomeningocele.

**Figure 16 diagnostics-11-01710-f016:**
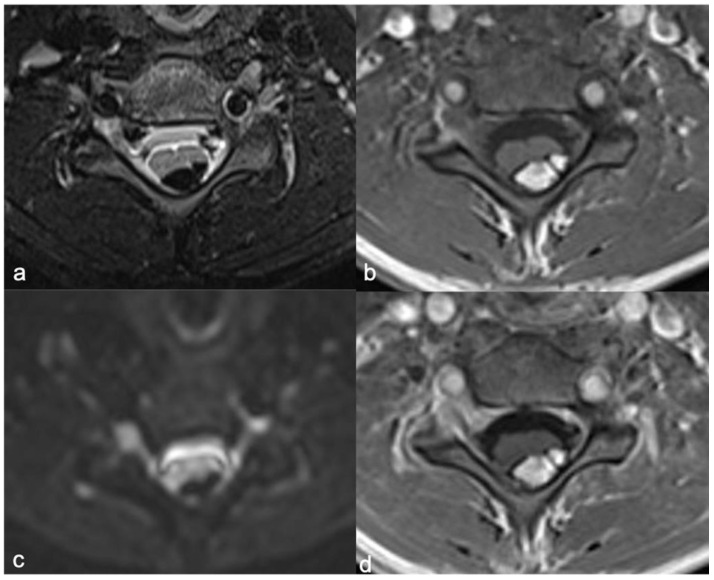
Intradural-extramedullary cervical expansive lesion in a ten-year-old child. Axial fat suppression T2-weighted (**a**), T1-weighted (**b**), DWI (**c**), fat suppression post-contrast T1-weighted images (**d**) demonstrate hyperintense lesion (T1 images), hypointense in fat suppression T2-w image, without diffusion restriction and enhancement. The diagnosis of lipoma is clear.

**Figure 17 diagnostics-11-01710-f017:**
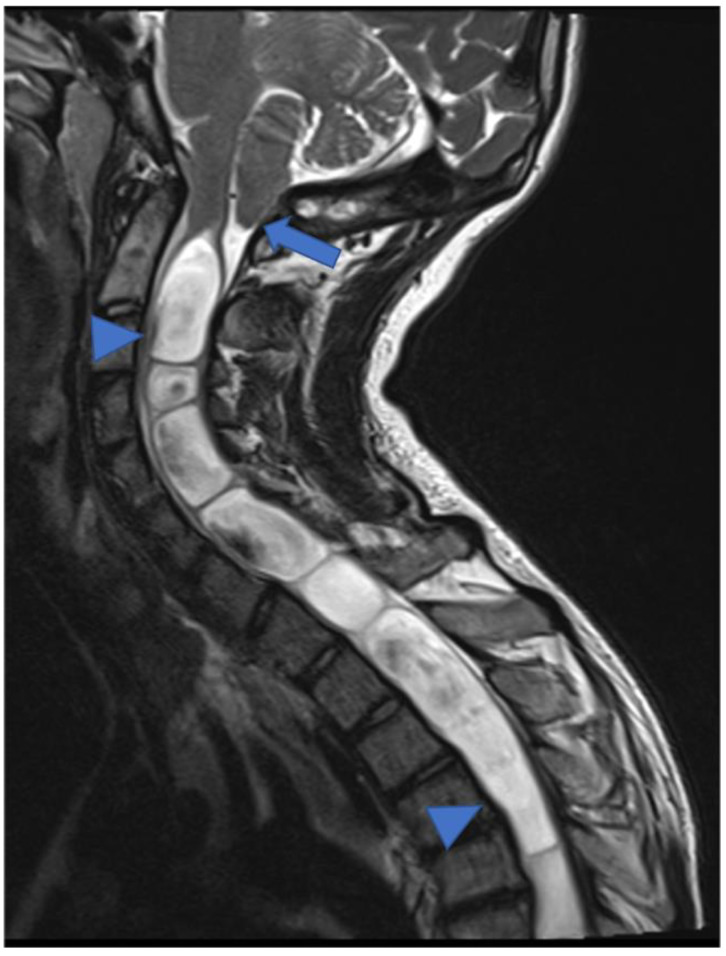
Cystic-like appearance of a tumor on the spinal cord in a three-year-old child. MRI was performed for headache. Sagittal T2-weighted image shows diffuse cervical–dorsal syringomyelia (arrowheads) not related to a neoplasm but to Chiari 1 syndrome. Migration of the cerebellar tonsils is evident (arrow).

**Figure 18 diagnostics-11-01710-f018:**
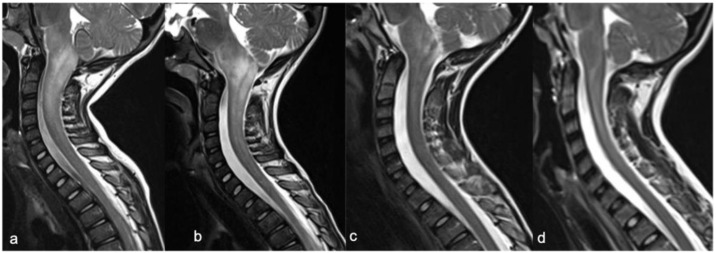
T2 hyperintense intramedullary cervical–bulbar area with an expansive aspect in a nine-year-old child. Sagittal T2-weighted images performed during surveillance (**a**–**d**)—the progressive spontaneous regression is evident. The definitive diagnosis is of UBOs in a patient with NF1.

**Figure 19 diagnostics-11-01710-f019:**
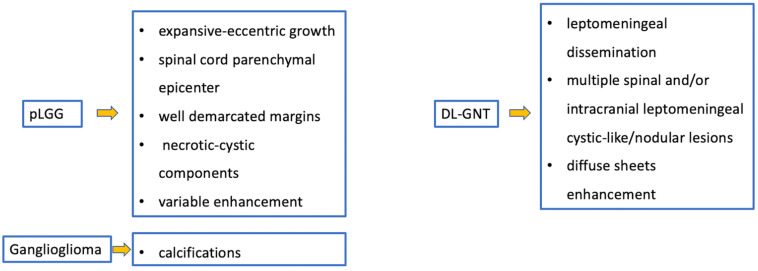
Gliomas and mixed neuronal–glial tumors. Imaging features and special “signs”.

**Figure 20 diagnostics-11-01710-f020:**
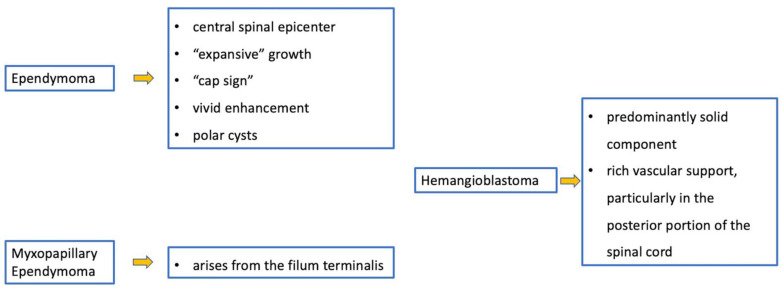
Ependymomas and hemangioblastomas. Imaging features and special “signs”.

**Figure 21 diagnostics-11-01710-f021:**
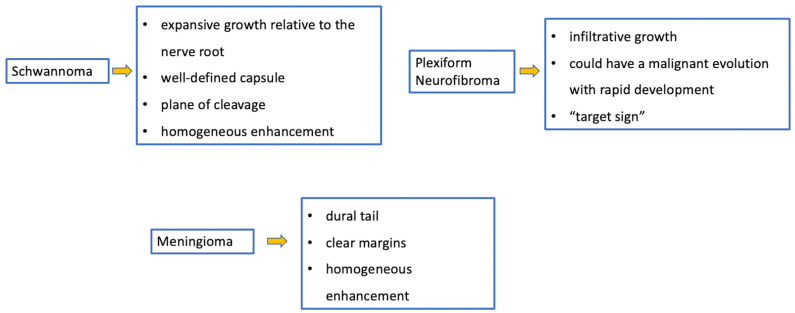
Meningiomas and tumors of the paraspinal nerves. Imaging features and special “signs”.

**Table 1 diagnostics-11-01710-t001:** Pediatric spinal tumors, location, and molecular data. * = the most common; / = no preferential location.

Tumors	Location *	Molecolar/Genetics
LGG	Cervico and thoracic tract	KIAA1549–BRAF fusion or BRAFV600E mutation; NF1 mutation
HGG	/	H3K27M
DL-GNT	Leptomeningeal dissemination	KIAA1549–BRAF, NTRK1/2/3, or TRIM33:RAF1 fusion
Ependymomas	Cervical/lumbo-sacral tract	RELA-/YAP1- fusion; nMyc amplification NF2 mutation
Hemangioblastomas	Variable epicenter	VHL mutation
Mesenchymal chondrosarcomas	Thoracic tract	HEY1/NCOA2 fusion
Meningiomas	/	NF2 mutation; SMARCE1, SMARCB1, or SUFU mutation
Schwannomas	Nerve sheaths	NF2 mutation; LZTR1 or SMARCB1 mutation
Plexiform neurofibromas	Nerve sheaths	NF1 mutation
Atypical teratoid/rhabdoid tumor	/	SMARCB1 mutation
Embryonal tumor with multilayered rosettes	/	C19MC amplification

## Data Availability

The data presented in this study are available on request from the corresponding author.

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
