# Peer review of "Intradural Pediatric Spinal Tumors: An Overview from Imaging to Novel Molecular Findings"

_diagnostics, 2021, doi:10.3390/diagnostics11091710_

Round 1
Reviewer 1 Report
This is a comprehensive and well written review on the topic of spinal tumors in children.
It has updated information re the WHO classification, molecular and methylation data and therapies.
It has a lot of information and figures relating to imaging characteristics of tumors but no pathology/histology at all.
Nevertheless, it is a nice review of the topic.
Author Response
This is a comprehensive and well written review on the topic of spinal tumors in children. It has updated information re the WHO classification, molecular and methylation data and therapies.It has a lot of information and figures relating to imaging characteristics of tumors but no pathology/histology at all.
- We thank a lot the reviewer; the topic of our review is to correlate the imaging characteristics with molecular findings; we have thought that histology figures are not mandatory to understand this correlation and the risk to overload the paper is too high
Nevertheless, it is a nice review of the topic.
Reviewer 2 Report
Overview – This is a review article for pediatric intradural spinal tumors. The abstract lays out the goals of this review which appear to be 3 fold: describe imaging of intradural tumors, the latest molecular findings, and genetic syndromes that may lead to intradural tumors. I would like to applaud the effort put forth in the article and think it has potential. The images are helpful and well placed.
I will disclose that I am not a radiologist but rather a pediatric neuro-oncologist, and therefore my review focused more on the accuracy of the clinical, pathologic, and molecular descriptions.
- General
- The entire manuscript needs to be carefully reviewed for spelling and grammar. I can appreciate the authors are likely writing many languages, but the grammar should be improved throughout and the issues with this really took away from what has the potential to be a helpful review article.
- Title
- The title suggests that they will be reviewing imaging and novel molecular findings. It is unclear from the title if the authors will be discussing the relation of these novel molecular findings to imaging findings. Later in the introduction, it becomes apparent that the authors will plan to investigate the connections.
- Abstract
- Minor – second last sentence states “pediatric spinal”, it appears there is a missing word after spinal.
- Introduction
- Minor – two-thirds and one-fourth do not makeup 100%. Should this be clarified?
- Minor – Suggest rewording the following sentence to improve flow. “The advancement of molecular findings in pediatric central nervous system tumors (CNS) provides a novel and additional information regarding the tumor subtypes, their biological aggressiveness and patient prognostic outcome, improving diagnosis and the therapeutic options thanks to new target drugs.”
- Minor – Suggest fixing the following sentence “Although these advances are less evident for pediatric spinal tumors in relation to their rarity respect compared to encephalic tumors, novel molecular findings are also described for the spinal neoplasmstic localization.”
- Minor – Suggest rewording of final introduction sentence. Are they just mentioning them or are they going to discuss these rare tumors in some sort of detail. Why is this interesting? Shouldn’t the authors just tell us they are going to discuss these tumors?
- Gliomas
- Minor – the authors state that pilocytic histological subtype is diagnosed in the first 5 years, but it can be diagnosed in other age groups. Suggest to put in that they “commonly” are diagnosed in the first five years.
- Minor – HGG in the spinal cord is uncommon for spinal tumors but is not “very rare”. Then in lines 89 and 90 the authors state the survival for this tumor but do not discuss this for other tumors in the same section. Perhaps better to be consistent and to put progression and survival data in similar orders within the text.
- Minor - in this case, it is important to distinguish between tumor and neoplastic edema. Can the authors share how this should be done? In my understanding, it is potentially complicated.
- Major – line 120, 121 - the cerebellar pLGGs [15] the ones spinal carrying KIAA1549-BRAF fusion seem to have a better prognosis compared to non-mutated [13]. This sentence should be reworded. Also, the claim that spinal tumors with the KIAA1549-BRAF fusion have a better prognosis than those that are non-mutated. Do the authors mean than those without the KIAA1549-BRAF fusion, should clarify which mutation they are talking about for the non-mutated group. Also in the cited study of citation #13 the difference for KIAA1549-BRAF fusions + vs – was not statistically significant so the prognostic statement may be misleading to the readers.
- Minor - The sentence on lines 121-123 should clarify that MEK inhibitors are used for BRAF fusion tumors while BRAF inhibitors are used for BRAF V600E tumors. Note that they are not BRAF V600E inhibitors, rather BRAF inhibitors. The BRAF inhibitors are often used in conjunction with MEK inhibitors for BRAF V600E mutant tumors.
- Minor – sentence lines 126-128 – the Histone H3.1 and H3.3 mutations a hallmark of the “diffuse midline glioma” rather than stating the midline pediatric high-grade glioma.
- Major – the authors should state the differences in what is low-grade vs high grade for the readers. As it stands the paper is a bit confusing as there is flipping between high-grade and low-grade but lacking definitions.
- Minor – sentence line 131-133. What does “spreading” mean? Do they mean metastasis?
- Minor clarify lines 134-135 – the specific aim should be “a primary management goal is…”
- Major – ganglioglioma is always low-grade unless there is anaplasia present. An anaplastic ganglioglioma is a high-grade tumor. The majority of gangliogliomas do not transform. The presence of both a BRAF V600E with either a TP53 or CDNK2A/B deletion makes transformation more likely. This needs to be clarified.
- Diffuse leptomeningeal glioneuronal tumor
- Minor – line 160-161. Should say “alterations leading to aberrant MAPK/ERK” rather than anomalous.
- Minor – line 165-166 “and loss of chromosomal arm 1p” this appears that this is only in the MC-1 group but 1p loss is also the MC-2 group. Suggest remove “and loss of chromosomal arm 1p” since this is implied in the definition of DL-GNT.
- Minor – ling 167 – “worst” should be “worse”
- Ependymoma
- Minor – line 171 – suggest replacing the word phacomatosis – really ependymomas are common in NF-2, perhaps just list out the tumor predisposition syndromes that lead to spinal ependymomas here. Or define phacomatosis.
- Minor – line 193 – instead of “these tumors” suggest stating “ependymoma”
- Major – line 194-198 – I would suggest these be removed since they do not apply to spinal ependymoma. Perhaps share more of the molecular items specifically with spinal ependymoma rather than the non-spinal ependymoma.
- Mesenchymal, non-meningothelial tumors
- Minor – line 208-209 should be reworded, hemangioblastomas are commonly found in patients with von Hippel-Lindau syndrome and findings of hemangioblastoma should prompt investigation for vHL.
- Major – As the authors describe other tumor types they should also describe the imaging features of the Mesenchymal chondrosarcoma rather than that they can occur in the spine. Also, I would suggest they include a figure to illustrate what this might look like.
- Meningioma and tumors of the paraspinal nerves
- Minor – line 230 and 231 – The authors are talking about pediatric spinal schwannomas but then state they make up 5% of all brain tumors – this should be “CNS tumors”? of spine tumors. The review is not talking about the brain so this should be updated.
- Minor – line 232 – NF2 should be spelled out in full the first time used.
- Major – the paragraph (line 230-244) flips between schwannomas more common in NF2 to plexiform neurofibromas more common in NF1. This is a bit confusing and the authors should be very explicit if they desire to keep both in the same paragraph. Both NF-1 and NF-2 might have a more thorough introduction here. The authors state in the introduction that they will discuss genetic predisposition syndromes and this is a good time to introduce these.
- Major – suggest that meningiomas occur often in NF2 (list percent), but can also occur sporadically (list percentage). In NF2 these meningiomas are not disseminated but more likely multiple primaries. There are many histologies and the authors only list clear cell, why? Is this most common pediatric meningioma histology?
- AT/RT
- Major – line 277-278, I think the authors are trying to say that SMARCB1 can occur in the germline which is a cancer predisposition syndrome, but cannot be sure.
- ETMR
- Minor - ETMR “encompasses” rather than “currently includes” as ETANTR and EBL and MEPL are no longer diagnostic entities.
- Genetic Syndromes
- Minor – line 293-297 – I am unclear about the purpose of this paragraph as it does not seem to relate to spinal tumors, except perhaps Li-Fraumeni on occasion.
- Minor – Line 301-302 – Please cite the new 2021 clinical guidelines for diagnosing NF-1 on clinical grounds rather than saying “such as.”
- Minor – Line 304 – They “can” be diffusely infiltrating astrocytomas” but this is truly an exception in pediatrics. Please highlight that this is uncommon.
- Minor – Line 307 does not make sense - the first in 57% of cases are located in the foramen
- Minor – Line 312 – “ed flag” I think the authors mean “red flag”
- Major – the paragraph 298 – 317 flips between glioma and plexiform neurofibroma. The authors need to guide the reader in a much clearer manner.
- Minor – Line 324 – should not be germinal but “germ-line”
- Minor – Line 340-341 is a duplicate from previous.
- Minor – Line 347 –symptoms may also be due to compression. May be better to state that the tumors can bleed and cause symptoms.
- Imaging technique and differential diagnoses
- Major – line 354-359 - Utility of whole-body MRI is certainly in question and prospective studies are needed. The reviewer feels the authors overstep here. See reply to “Whole-Body MRI Screening in Children With Li-Fraumeni and Other Cancer-Predisposition Syndromes.” Certainly, we do not order whole-body MRI for NF-1 or NF-2. Perhaps in NF-1, a PET scan may be warranted for plexiform neurofibroma if concern for transformation.
- Minor – line 363 – what does “spreading” mean is metastases?
- Target therapies should be ”targeted therapies”
- Major – Line 483 – surgical resection is the mainstay when a tumor is There are many tumors that may be “observed” only.
- Minor – line 487-488 – It is not the rare incidence of spinal cord neoplasms in and of itself but also “heterogeneous histologies”
- Minor – line 493 – it is not BRAF V600 inhibitors – but BRAF inhibitors.
- Minor – line 496-497 – 44% response rate
- Major – line 504-506 – IDH1 mutations are rare in children. Suggest to note this or exclude.
- Minor – line 507 Even though (non-capital letter).
- Major – line 507-511 – Targeted therapies in VHL have been disappointing to date. Surgery remains a mainstay, sometimes radiation therapy. Unclear why drug names are listed here but not in other targeted therapy paragraphs.
- Major – line 512-514 Schwannomas have not been shown clinically to benefit patients and the data is pre-clinical – needs to be clarified.
- Conclusion
- Minor – suggest removing “excluding other diseases such as acute transverse myelopathy (ATM) and leptomeningeal infections”
- Minor DWI may help with detecting metastases but is certainly not the only sequence to aid in assessing metastases.

Author Response
Overview – This is a review article for pediatric intradural spinal tumors. The abstract lays out the goals of this review which appear to be 3 fold: describe imaging of intradural tumors, the latest molecular findings, and genetic syndromes that may lead to intradural tumors. I would like to applaud the effort put forth in the article and think it has potential. The images are helpful and well placed.
I will disclose that I am not a radiologist but rather a pediatric neuro-oncologist, and therefore my review focused more on the accuracy of the clinical, pathologic, and molecular descriptions.
- General
- The entire manuscript needs to be carefully reviewed for spelling and grammar. I can appreciate the authors are likely writing many languages, but the grammar should be improved throughout and the issues with this really took away from what has the potential to be a helpful review article. The text was revised by a native English speaker
- Title
- The title suggests that they will be reviewing imaging and novel molecular findings. It is unclear from the title if the authors will be discussing the relation of these novel molecular findings to imaging findings. Later in the introduction, it becomes apparent that the authors will plan to investigate the connections.
- Abstract
- Minor – second last sentence states “pediatric spinal”, it appears there is a missing word after spinal. Word "tumors" added
- Introduction
- Minor – two-thirds and one-fourth do not makeup 100%. Should this be clarified? Clarified: "spinal neoplasms are divided into extradural (two-thirds of cases), intradural-extramedullary and intramedullary (one-third of cases) lesions"
- Minor – Suggest rewording the following sentence to improve flow. “The advancement of molecular findings in pediatric central nervous system tumors (CNS) provides a novel and additional information regarding the tumor subtypes, their biological aggressiveness and patient prognostic outcome, improving diagnosis and the therapeutic options thanks to new target drugs.”
The advancement of molecular findings in pediatric central nervous system tumors (CNS) provides additional informations regarding the tumor subtypes and their biological behavior and patient outcome. These novel findings help to reach detailed diagnosis obtaining new therapeutic options by targeted drugs. These molecular advances are described also for spinal neoplasms.
- Minor – Suggest fixing the following sentence “Although these advances are less evident for pediatric spinal tumors in relation to their rarity respect compared to encephalic tumors, novel molecular findings are also described for the spinal neoplasmstic localization.” Previous sentence better explained
- Minor – Suggest rewording of final introduction sentence. Are they just mentioning them or are they going to discuss these rare tumors in some sort of detail. Why is this interesting? For their typical radiological aspect .Shouldn’t the authors just tell us they are going to discuss these tumors? Among these very rare tumors, there will be a brief mentioning text about them
- Gliomas
- Minor – the authors state that pilocytic histological subtype is diagnosed in the first 5 years, but it can be diagnosed in other age groups. Suggest to put in that they “commonly” are diagnosed in the first five years. Suggestion accepted
- Minor – HGG in the spinal cord is uncommon for spinal tumors but is not “very rare”. Then in lines 89 and 90 the authors state the survival for this tumor but do not discuss this for other tumors in the same section. Perhaps better to be consistent and to put progression and survival data in similar orders within the text. Suggestion accepted
- Minor - in this case, it is important to distinguish between tumor and neoplastic edema. Can the authors share how this should be done? In my understanding, it is potentially complicated. "The swelling spine appearance could extend trough several vertebral segments (usually <4) [7] up to being holocord: in this case it is important (but often complicated and not ever feaseble)to distinguish between tumor and neoplastic edema" There is not a specific way to distinguish it
- Major – line 120, 121 - the cerebellar pLGGs [15] the ones spinal carrying KIAA1549-BRAF fusion seem to have a better prognosis compared to non-mutated [13]. This sentence should be reworded. Also, the claim that spinal tumors with the KIAA1549-BRAF fusion have a better prognosis than those that are non-mutated. Do the authors mean than those without the KIAA1549-BRAF fusion, should clarify which mutation they are talking about for the non-mutated group. Also in the cited study of citation #13 the difference for KIAA1549-BRAF fusions + vs – was not statistically significant so the prognostic statement may be misleading to the readers.“ It is largely investigated and demonstrated that KIAA1549-BRAF fused cerebellar pLGGs have a better prognosis compared to the ones not carrying the fusion; the same evidence is described for the spinal low grade gliomas [15] . However, the association between KIAA1549 – BRAF fusion and outcome is not yet validated, as according to some studies, this molecular alteration could not predict the prognosis [18]” In this way we have clarified that it is a fusion and not the mutation and we added the right bibliography
- Minor - The sentence on lines 121-123 should clarify that MEK inhibitors are used for BRAF fusion tumors while BRAF inhibitors are used for BRAF V600E tumors. Note that they are not BRAF V600E inhibitors, rather BRAF inhibitors. The BRAF inhibitors are often used in conjunction with MEK inhibitors for BRAF V600E mutant tumors. Thanks to reviewer, we have changed the sentence
- Minor – sentence lines 126-128 – the Histone H3.1 and H3.3 mutations a hallmark of the “diffuse midline glioma” rather than stating the midline pediatric high-grade glioma. Reworded
- Major – the authors should state the differences in what is low-grade vs high grade for the readers. As it stands the paper is a bit confusing as there is flipping between high-grade and low-grade but lacking definitions. Added the WHO definition for both the gliomas
- Minor – sentence line 131-133. What does “spreading” mean? Do they mean metastasis? Yes, changed with metastasis
- Minor clarify lines 134-135 – the specific aim should be “a primary management goal is…” Clarified in this way: “A primary management goal for pediatric spinal tumors is to extend long-term follow-up because progression is possible even 10-20 years after diagnosis; for this reason is important to enlarge tumors’ knowledge [18]. “
- Major – ganglioglioma is always low-grade unless there is anaplasia present. An anaplastic ganglioglioma is a high-grade tumor. The majority of gangliogliomas do not transform. The presence of both a BRAF V600E with either a TP53 or CDNK2A/B deletion makes transformation more likely. This needs to be clarified.
Added bibliography and clarified the sentence.These neoplasms, usually low-grade, may be characterized by local recurrence, but very uncommonly by the risk of malignant evolution (usually when both a BRAFV600E and a TP53 or CDNK2A/B deletion are present) [21][22].
- Diffuse leptomeningeal glioneuronal tumor
- Minor – line 160-161. Should say “alterations leading to aberrant MAPK/ERK” rather than anomalous. Edited
- Minor – line 165-166 “and loss of chromosomal arm 1p” this appears that this is only in the MC-1 group but 1p loss is also the MC-2 group. Suggest remove “and loss of chromosomal arm 1p” since this is implied in the definition of DL-GNT. Suggestion accepted
- Minor – ling 167 – “worst” should be “worse” Corrected word
- Ependymoma
- Minor – line 171 – suggest replacing the word phacomatosis – really ependymomas are common in NF-2, perhaps just list out the tumor predisposition syndromes that lead to spinal ependymomas here. Or define phacomatosis. Definition of phacomatosis as genetic neurocutaneous disorders.
- Minor – line 193 – instead of “these tumors” suggest stating “ependymoma” Reworded
- Major – line 194-198 – I would suggest these be removed since they do not apply to spinal ependymoma. Perhaps share more of the molecular items specifically with spinal ependymoma rather than the non-spinal ependymoma. Added some molecular findings of spinal ependymomas and deleted the ones referring to brain ependymomas
- Mesenchymal, non-meningothelial tumors
- Minor – line 208-209 should be reworded, hemangioblastomas are commonly found in patients with von Hippel-Lindau syndrome and findings of hemangioblastoma should prompt investigation for vHL. Reworded
- Major – As the authors describe other tumor types they should also describe the imaging features of the Mesenchymal chondrosarcoma rather than that they can occur in the spine. Also, I would suggest they include a figure to illustrate what this might look like “Due to the different site of origin, mesenchymal chondrosarcoma can have different imaging characteristics. Tumors may show inhomogoneous T2 hyperintensity and inhomogeneous enhancement but these features are related to the specific case”
- Meningioma and tumors of the paraspinal nerves
- Minor – line 230 and 231 – The authors are talking about pediatric spinal schwannomas but then state they make up 5% of all brain tumors – this should be “CNS tumors”? of spine tumors. The review is not talking about the brain so this should be updated. Edited in :They account for 0.3% of intraspinal tumors [42]
- Minor – line 232 – NF2 should be spelled out in full the first time used. Done
- Major – the paragraph (line 230-244) flips between schwannomas more common in NF2 to plexiform neurofibromas more common in NF1. This is a bit confusing and the authors should be very explicit if they desire to keep both in the same paragraph. Both NF-1 and NF-2 might have a more thorough introduction here. The authors state in the introduction that they will discuss genetic predisposition syndromes and this is a good time to introduce these. We have preferred to let genetic discussion in the other paragraph
- Major – suggest that meningiomas occur often in NF2 (list percent), but can also occur sporadically (list percentage). In NF2 these meningiomas are not disseminated but more likely multiple primaries. There are many histologies and the authors only list clear cell, why? Is this most common pediatric meningioma histology? Specified, as suggested, “The “clear cell meningioma (CCM)” (WHO grade II) is a typical pediatric/juvenile spinal meningioma and represents the most common histological subtype of sporadic pediatric spinal meningioma”
- AT/RT
- Major – line 277-278, I think the authors are trying to say that SMARCB1 can occur in the germline which is a cancer predisposition syndrome, but cannot be sure. Exactly, so we changed the sentence
- ETMR
- Minor - ETMR “encompasses” rather than “currently includes” as ETANTR and EBL and MEPL are no longer diagnostic entities. Edited
- Genetic Syndromes
- Minor – line 293-297 – I am unclear about the purpose of this paragraph as it does not seem to relate to spinal tumors, except perhaps Li-Fraumeni on occasion. Better explained that also NF1 and NF2 are correlated to spinal tumors
- Minor – Line 301-302 – Please cite the new 2021 clinical guidelines for diagnosing NF-1 on clinical grounds rather than saying “such as.” 2021 guidelines inserted
- Minor – Line 304 – They “can” be diffusely infiltrating astrocytomas” but this is truly an exception in pediatrics. Please highlight that this is uncommon.Highlighted
- Minor – Line 307 does not make sense - the first in 57% of cases are located in the foramen Deleted the sentence
- Minor – Line 312 – “ed flag” I think the authors mean “red flag” Edited
- Major – the paragraph 298 – 317 flips between glioma and plexiform neurofibroma. The authors need to guide the reader in a much clearer manner. Edited the paragraph
- Minor – Line 324 – should not be germinal but “germ-line” Edited
- Minor – Line 340-341 is a duplicate from previous. We thank the reviewer and modified the sentence but thinking that it can better explain
- Minor – Line 347 –symptoms may also be due to compression. May be better to state that the tumors can bleed and cause symptoms. "New neurological symptoms/deficits, pain, changes in the growth and consistency, compression of the locoregional areas, bleeding of the neurofibroma are the so-called "red flag" symptoms"
- Imaging technique and differential diagnoses
- Major – line 354-359 - Utility of whole-body MRI is certainly in question and prospective studies are needed. The reviewer feels the authors overstep here. See reply to “Whole-Body MRI Screening in Children With Li-Fraumeni and Other Cancer-Predisposition Syndromes.” Certainly, we do not order whole-body MRI for NF-1 or NF-2. Perhaps in NF-1, a PET scan may be warranted for plexiform neurofibroma if concern for transformation. Specified that is a controversial question, still open baut added bibliography. “ Cancer predisposition syndromes (CPSs) differ from each other in relation to the site of onset of the neoplasm, type of neoplasm and involvement (or not) of the CNS. In this setting, whole-body MRI is the preferred imaging modality for surveillance of pediatric patients with CPSs, and the growing literature supports its importance in presymptomatic cancer detection, but further studies are needed and the question is still open. . Nevertheless , evaluation and follow-up of children with CNS tumor, not only in the CPSs, is based on brain and spine MRI.”
- Minor – line 363 – what does “spreading” mean is metastases? Yes, clarified
- Target therapies should be ”targeted therapies” Edited
- Major – Line 483 – surgical resection is the mainstay when a tumor is There are many tumors that may be “observed” only. Better explained with the sentence Surgical resection is the mainstay of treatment for those spinal tumors not suitable for only observational follow up; surgery is undertaken when feasible and safe to perform with curative intent
- Minor – line 487-488 – It is not the rare incidence of spinal cord neoplasms in and of itself but also “heterogeneous histologies” Clarified
- Minor – line 493 – it is not BRAF V600 inhibitors – but BRAF inhibitors. Modified
- Minor – line 496-497 – 44% response rate Added the word
- Major – line 504-506 – IDH1 mutations are rare in children. Suggest to note this or exclude. Added" rarely in pediatric age"
- Minor – line 507 Even though (non-capital letter). Modified
- Major – line 507-511 – Targeted therapies in VHL have been disappointing to date. Surgery remains a mainstay, sometimes radiation therapy. Unclear why drug names are listed here but not in other targeted therapy paragraphs. Names of drugs put in this paragraph because it is the only one regarding therapy
- Major – line 512-514 Schwannomas have not been shown clinically to benefit patients and the data is pre-clinical – needs to be clarified. Specified that are preclinical data
- Conclusion
- Minor – suggest removing “excluding other diseases such as acute transverse myelopathy (ATM) and leptomeningeal infections” Suggestion accepted
- Minor DWI may help with detecting metastases but is certainly not the only sequence to aid in assessing metastases. Modified : An MRI evaluation includes pre- and post-contrast T1- weighted sequences, T2-weighted sequences, MR "cisternography" sequences; DWI plays an important role in the assessment and detection of metastases
Reviewer 3 Report
This review is good and the content is new and fruitful. HOwever, the whole article needs an English native speaker to have a complete revision in English.
line 59: Our aim in this review is to describe the imaging"findings" of the most frequent intradural
line 71 five years of life while fibrillary subtype are found around "at" age 10, accounting for 75%
Line 76: The swollen swelling spine appearance could extend for through several vertebral segments
Line 79 (i) solid (40% of cases), (ii) necrotic-cystic (60% of cases) or (iii) nodular-cystic (fig 2)"." "T"herefore, it could appear (i) iso/hypointense in T1 and hyperintense in T2 or (ii) hyper
Line 81Absent in 30% of cases [9],???
Line 82: iable and depends on the different components"." "H"owever it is less evident than that of
line 116: Please do not use abbreviation. For instance: BRAF: fullname V-raf murine sarcoma viral oncogene homolog B.
line 124: the association between KIAA1549 – BRAF fusion and outcome (?) is not yet validated,
line 127: considered "as" a hallmark of midline pediatric high-grade (both cerebral and spinal)
line 129 [19]"." "T"his is important to highlight the pathological heterogeneity in gliomas and the
line 133-135: A specific aim for pediatric spinal tumors is to extend long-term follow-up because progression is possible even 10- 20 years after diagnosis (please revise this sentence with an English native speaker, because this sentence is very confusing and not clear.
line 139-140: related to slow grows growth possible
line 160: The typical molecular landscape of DL-GNT concerns(?) an anomalous MAPK/ERK path
line 212: fullname, please
line 251-257: , for example SMARCE1 (SWI/SNF Related, Matrix Associated, Actin Dependent Regulator Of Chromatin, Subfamily E, Member 1), fullname, please
line 348: To conclude: In conclusion
Author Response
This review is good and the content is new and fruitful. HOwever, the whole article needs an English native speaker to have a complete revision in English.
line 59: Our aim in this review is to describe the imaging"findings" of the most frequent intradural Word added
line 71 five years of life while fibrillary subtype are found around "at" age 10, accounting for 75% Word added
Line 76: The swollen swelling spine appearance could extend for through several vertebral segments Suggestion inserted
Line 79 (i) solid (40% of cases), (ii) necrotic-cystic (60% of cases) or (iii) nodular-cystic (fig 2)"." "T"herefore, it could appear (i) iso/hypointense in T1 and hyperintense in T2 or (ii) hyper Suggestion accepted
Line 81Absent in 30% of cases [9],??? Edited with “contrast enhancement”
Line 82: iable and depends on the different components"." "H"owever it is less evident than that of Corrected
line 116: Please do not use abbreviation. For instance: BRAF: fullname V-raf murine sarcoma viral oncogene homolog B. Edited
line 124: the association between KIAA1549 – BRAF fusion and outcome (?) is not yet validated:
Better prognosis is described; added also bibliography “It is largely investigated and demonstrated that KIAA1549-BRAF fused cerebellar pLGGs have a better prognosis compared to the ones not carrying the fusion; the same evidence is described for the spinal low grade gliomas [15] .”
line 127: considered "as" a hallmark of midline pediatric high-grade (both cerebral and spinal) Corrected
line 129 [19]"." "T"his is important to highlight the pathological heterogeneity in gliomas and the Corrected
line 133-135: A specific aim for pediatric spinal tumors is to extend long-term follow-up because progression is possible even 10- 20 years after diagnosis (please revise this sentence with an English native speaker, because this sentence is very confusing and not clear. Modified sentence : A primary management goal for pediatric spinal tumors is to extend long-term follow-up because progression is possible even 10-20 years after diagnosis; for this reason is important to enlarge tumors’ knowledge
line 139-140: related to slow grows growth possible Corrected
line 160: The typical molecular landscape of DL-GNT concerns(?) an anomalous MAPK/ERK path Reworded with “is characterized by”
line 212: fullname, please Added
line 251-257: , for example SMARCE1 (SWI/SNF Related, Matrix Associated, Actin Dependent Regulator Of Chromatin, Subfamily E, Member 1), fullname, please Modified
line 348: To conclude: In conclusion Edited
Round 2
Reviewer 2 Report
I believe the authors have adequately addressed my concerns. I appreciate their patience and effort in finalizing the last reviews which I feel aids in the manuscript flow.
Author Response
Thank to the reviewer for the comment about the last version edited after his suggestions.
Reviewer 3 Report
The authors have completed all my questions. I recommend this work should be accepted.
Author Response
Thanks to the reviewer about this comment and the prevue suggestions